# Memory Never Fades: Boosting Long Context Processing with Global Memory-Enhanced Retrieval Augmentation

## Abstract

Processing long contexts presents a significant challenge for large language models (LLMs). While recent advancements allow LLMs to handle much longer contexts than before (e.g., 32K or 128K tokens), it is computationally expensive and can still be insufficient for many applications. Retrieval-Augmented Generation (RAG) is considered as a promising strategy to address this problem. However, conventional RAG methods face inherent limitations because of two underlying requirements: 1) explicitly-stated queries, and 2) well-structured knowledge. These conditions, however, do not hold in general long-context processing tasks.

In this work, we propose **HawkRAG**[1], a novel RAG framework empowered by global memory-augmented retrieval. HawkRAG features a dual-system architecture. First, it employs a *light but long-range system* to create a global memory of the long context. Once a task is presented, it generates draft answers, providing useful clues for the retrieval tools to locate relevant information within the long context. Second, it leverages an *expensive but expressive system*, which generates the final answer based on the retrieved information. Building upon this fundamental framework, we realize the memory module in the form of KV compression, and reinforce its memorization and cluing capacity from the Generation quality's Feedback (*a.k.a.* RLGF). In our experiments, HawkRAG achieves superior performances across a variety of long-context evaluation tasks, not only complex scenarios where traditional RAG methods struggle, but also simpler ones where RAG is typically applied. Our source code is available at *this anonymous repository*.

## CCS Concepts

• **Computing methodologies** → **Natural language generation**.

## Keywords

Retrieval-Augmented Generation, Long Context Processing

**ACM Reference Format:**
Anonymous Author(s). 2018. Memory Never Fades: Boosting Long Context Processing with Global Memory-Enhanced Retrieval Augmentation. In *Proceedings of Make sure to enter the correct conference title from your rights confirmation emai (Conference acronym 'XX)*. ACM, New York, NY, USA, 14 pages. https://doi.org/XXXXXXX.XXXXXXX

[1]The name HawkRAG is inspired by the way a hawk glides high in the sky to observe the land, allowing it to spot and target prey with precision from a broad vantage point.

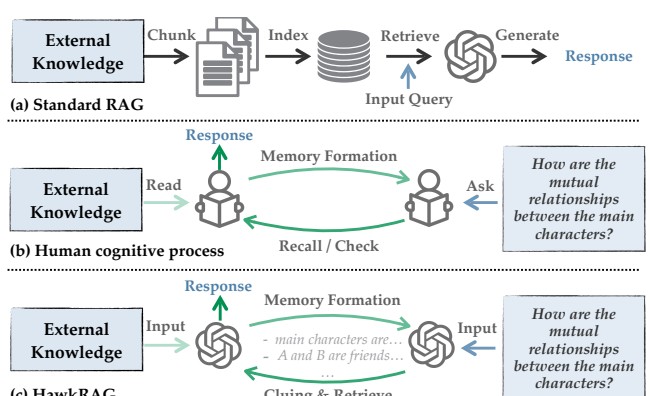

**Figure 1: Comparison of HawkRAG with Standard RAG and human cognition of a long document. Figure (a) shows standard RAG, where retrieval and generation take place in a sequential pipeline. Figure (b) illustrates how human tackle a task about the document: 1. going-through the document and forming the memory, 2. thinking about the clues to the presented task (i.e., recalling), checking the document for needed details (i.e., retrieving), 3. making response to the task based on the memory-enhanced retrieval result. Inspired by human cognition process, Figure (c) demonstrates HawkRAG, which creates a global memory of the long-context, recalling useful clues based on memory, retrieving needed information based on the clues to generate a high-quality response.**

## 1 Introduction

Large language models (LLMs) need to process long contexts in many real-world scenarios, such as long-document QA and summarization [4, 55]. While some recent LLMs can handle much longer contexts than before (e.g., Mistral-32K, Phi-128K) [1, 23], they can still be insufficient for certain applications. Meanwhile, it's computationally expensive to process long contexts directly due to the considerable costs on inference time and GPU memory [11].

Retrieval-Augmented Generation (RAG) is widely regarded as a promising strategy for addressing long-context processing challenges [16, 22]. RAG allows LLMs to complete tasks more cost-effectively by focusing only on the relevant parts retrieved from the long input context [51, 56]. However, traditional RAG methods face inherent limitations when applied to general long-context tasks, due to two key constraints. First, the search intent must be explicitly expressed (or easily clarified through query rewriting) [6, 56]. Second, the external dataset must be well-structured for effective encoding and indexing (e.g., Wikipedia passages) [35, 36]. Unfortunately, neither of these conditions are typically met in general long-context tasks. On one hand, there may be no clear search intent (e.g., summarizing the main characters in a book, or clarifying the relationships between characters) [13, 40]. On the other hand,

the input context is often unstructured (e.g., a 100-page text file, or multi-year financial reports), making it difficult to partition, encode, and index in a straightforward manner [39, 41, 56].

Human cognition of a long document, unlike standard RAG, is significantly more effective (as shown in Figure 1). When a person is presented with a long document, they first skim through it to form a global memory of its high-level information. When tasked with a document understanding question—such as "What are the mutual relationships between the main characters?"—the person recalls useful clues from their memory and uses these clues to locate specific details within the document. Based on the retrieved information, they can then generate a high-quality response to the task [2].

Inspired by human cognitive process, we propose **HawkRAG**, a novel framework for long-context processing on top of global-memory enhanced retrieval augmentation. HawkRAG features a dual-system architecture: a light but long-range system to realize the memory module and a heavy but expressive system to generate the final answer. For each presented task, HawkRAG prompts its memory module to generate retrieval clues. These clues are essentially drafted answers based on the compact memory. While these clues may contain some inaccuracies or lack details, they effectively reveal the underlying information needs of the task and can be directly linked to the source information. By using these clues as queries, HawkRAG can effectively retrieve the necessary knowledge from the external knowledge base.

The memory module is core of HawkRAG. It is expected to be 1) *length-scalable*: handling long-contexts in a cost-effective way, 2) *retentive*: memorizing the crucial information within long-contexts, and 3) *instructive*: generating useful clues for the presented task. Therefore, we introduce the following techniques to optimize its performance. First, we realize the the memory module in the form of a **KV-compressible LLM** with configurable compression rates. This structure is able to flexibly support a wide range of context lengths and can be optimized in an end-to-end manner. Second, we design a novel algorithm which learns to reinforce the memory module's memorization and cluing capacity from the generation quality's feedback (*a.k.a.* **RLGF**). That is, 1) the generated clues are positively rewarded if it can support the generation of high-quality answer, and 2) the memory module is reinforced to generate the positively rewarded clues.

We perform comprehensive experimental studies to evaluate HawkRAG. In our experiment, we leverage a variety of datasets from two popular long-context benchmarks: LongBench [4] and InfiniteBench [55]. The two benchmarks contain both QA-style tasks, e.g., HotPotQA, NarrativeQA, which are well-suited for traditional RAG methods, and non-QA tasks, like government report summarization, which are unfavorable to traditional RAG methods. We also curate a general long-document understanding benchmark, containing general tasks related to long documents from 20 diverse domains, such as law, finance, physics, and programming etc. Our experiment results lead to a series of critical insights. *Firstly*, HawkRAG not only achieves notable advantages in both non-QA tasks where traditional RAG methods struggle, but also QA-style tasks where traditional RAG methods are usually applied. *Secondly*, HawkRAG outperforms advanced retrieval and RAG methods which are proposed recently, such as HyDE [15],

RQ-RAG [6], and GraphRAG [13]. *Thirdly*, HawkRAG even outperforms the direct-applied long LLMs and some context-extended methods, which can fully cover the input contexts [1, 24]. Finally, HawkRAG exhibits competitive efficiency in terms of inference speed and memory cost. To summarize, the contributions of our work are highlighted by the following points.

- We propose HawkRAG for long-context processing tasks based on global-memory enhanced retrieval augmentation.
- We design a suite of architecture and optimization algorithm, enabling the memory module to be length-scalable, retentive, and instructive for long-context tasks.
- We empirically demonstrate that HawkRAG generalizes beyond traditional QA tasks to effectively handle both non-QA tasks and complex QA tasks, expanding RAG's applicability to a broader range of scenarios.

## 2 Method

In this section, we begin by introducing task background and presenting the overall framework of HawkRAG, followed by a detailed exploration of HawkRAG's technical designs.

### 2.1 Background

The generation process of a LLM $\Theta(\cdot)$ can be succinctly represented as $Y = \Theta(q \mid \theta)$, where $q$ denotes the input query, $Y$ is the generated response, and $\theta$ represents the model's parameters, which store the knowledge learned from the training corpus. Since the training corpus typically consists of publicly available web data up to a certain cutoff point, LLMs face challenges when handling tasks that require up-to-date or domain-specific information. A common and effective solution to this problem is to incorporate an external knowledge base $C$ into the input, which can be formulated as $Y = \Theta(q, C \mid \theta)$, allowing for more accurate responses. In practice, the external knowledge base $C$ can be substantially large, often exceeding the LLM's context size, leading to the *long-context issue*, as shown in the top of Figure 2(a). In the following, we refer to the external knowledge base $C$ as the long input context.

A straightforward idea to address the *long-context issue* is to employ LLMs with long-context processing ability. However, despite recent advancements in increasing context lengths, handling very long contexts remains infeasible for most LLMs, often resulting in incomplete answers as the context is truncated. Besides, RAG has emerged as a widely adopted solution to enable LLMs to effectively handle the *long-context issue*. RAG allows LLMs to retrieve and leverage only relevant information from the long context. A standard RAG system typically consists of two components: a generation model, $\Theta(\cdot)$, and a retrieval model, $\Gamma(\cdot)$. Given an input query $q$, the retrieval model $\Gamma$ first identifies the relevant evidence $E$ from the long context $C$. This retrieved evidence is then passed to the generation model $\Theta$, which utilizes it to produce the final response $Y$. Formally, this process can be described as:

$$Y = \Theta(q, E \mid \theta), \quad E = \Gamma(q, C). \quad (1)$$

In an ideal retrieval setting, the query $q$ serves as a piece of text that is representative of the expected evidence [34], allowing the retriever to easily locate the relevant evidence $E$. However, as shown in the bottom of Figure 2(a), in many practical scenarios,

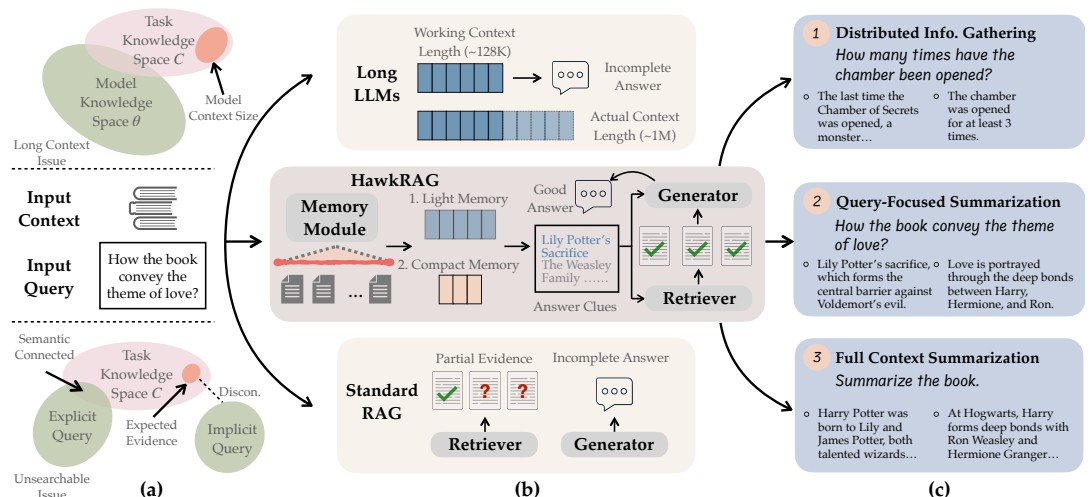

**Figure 2: Illustration of (a) task background, (b) framework comparison, and (c) application scenarios. When processing long inputs like the entire Harry Potter series, most LLMs struggle with million-token contexts. Standard RAG methods also face challenges with queries unsuitable for direct searching. HawkRAG overcomes these limitations by constructing a global memory that generates clues, guiding the retrieval of relevant evidence and enabling more accurate and comprehensive answers.**

the input query $q$ often carries implicit information-seeking intents that are not semantically aligned with the expected text evidence. As a result, standard retrievers, which typically rely on lexical or semantic matching, may struggle to accurately retrieve the expected evidence, leading to performance degradation in RAG systems. This issue underscores the need for an advanced RAG framework to bridge the semantic gap frequently encountered in such situations.

## 2.2 HawkRAG

In this paper, we propose HawkRAG, which leverages a memory model $\Theta_{\text{mem}}(\cdot)$ to learn and store the long context $C$, forming a global memory denoted as $\theta_{\text{mem}}$. When a query or task instruction $q$ is presented, HawkRAG prompts the memory model to generate draft answers $y$, which serve as a set of answer clues. These clues guide the retrieval of accurate and comprehensive evidence $E$ from the long context $C$. Subsequently, the final answer $Y$ is generated using the retrieved evidence text $E$. This process is defined as:

$$Y = \Theta(q, E \mid \theta), \quad E = \Gamma(y, C), \quad y = \Theta_{\text{mem}}(q \mid \theta_{\text{mem}}). \quad (2)$$

HawkRAG is illustrated in the middle of Figure 2(b).

To facilitate understanding, we illustrate HawkRAG framework with pseudo-code in Algorithm 1.

Specifically, in  line 1 , HawkRAG begins by receiving a long input context $C$, which is combined with auxiliary text (e.g., prompts), referred to as the input sequence $\mathcal{X}$. HawkRAG's memory model then processes $\mathcal{X}$ to form a global memory representation, denoted as $\theta_{\text{mem}}$ in  line 2  (see Section 2.3 for details on the memory model). This memory representation, $\theta_{\text{mem}}$, encapsulates the high-level semantics of the entire long context from a global perspective. In practice, the memory can be offloaded for efficient reuse in future tasks. In  line 6 , when a query $q$ is presented, the global memory $\theta_{\text{mem}}$ is used to generate task-specific clues, denoted as $y$. These clues serve to outline the expected answer $Y$, effectively bridging

---

**Algorithm 1** HawkRAG Framework

1:  **Input**: long context $C$, memory model $\Theta_{\text{mem}}(\cdot)$
2:  **Memory Formation**: Generate global memory $\theta_{\text{mem}} = \Theta_{\text{mem}}(\mathcal{X})$, $\mathcal{X} = C$ + auxiliary text
3:  **Input**: queries $\{q_1, \ldots, q_n\}$, generator $\Theta(\cdot)$, retriever $\Gamma(\cdot)$
4:  **Initialize**: answer set $\mathcal{Y} \leftarrow \{\}$
5:  **for** each query $q_i \in \{q_1, \ldots, q_n\}$ **do**
6:     $y_i = \Theta_{\text{mem}}(q_i \mid \theta_{\text{mem}})$ # Generate draft answer clues for $q_i$
7:     $E_i = \Gamma(y_i, C)$ # Retrieve relevant evidence based on the clues
8:     $Y_i = \Theta(q_i, E_i \mid \theta)$ # Generate the final answer for $q_i$
9:     $\mathcal{Y} \leftarrow \mathcal{Y} \cup \{Y_i\}$ # Add final answer to the answer set
10: **end for**
11: **Optional - Memory Offload**: Save global memory $\theta_{\text{mem}}$ to disk for future reuse
12: **Return**: answer set $\mathcal{Y}$

---

the gap between the raw input context and the ground-truth answer. Based on these memory-generated clues, HawkRAG's retriever is employed to locate precise evidence text $E$ within the long input context, as shown in  line 7 . Using the retrieved evidence text $E$ along with the input query $q$, HawkRAG's generator produces the final response $Y$, shown in  line 8 . By default, HawkRAG utilizes the memory model's underlying LLM as the generator to ensure parameter efficiency.

*Application Scenario.* HawkRAG can adapt to a variety of application scenarios and determine how to generate appropriate clues based on the specific type of long-context task presented. In Figure 2(c), we illustrate three scenarios that are particularly challenging for standard RAG but well-suited for HawkRAG. First, in a question-answering task where the query requires gathering distributed information, HawkRAG generates answer clues $y$ that include intermediary reasoning steps, such as creating more explicit surrogate queries and retrieving relevant evidence from the

long context to support the final answer. Second, in query-focused summarization tasks, the queries are inherently unsearchable, as the target information must be aggregated from the entire context rather than isolated segments. Since HawkRAG has already comprehended the entire long context, it can recall multiple query-related evidence clues, enabling more effective information retrieval and synthesis. Third, for tasks without explicit queries, such as text summarization, the draft answer may consist of key points or concepts extracted from the context, which are essential for constructing a coherent and accurate summary.

## 2.3 Memory Module

As discussed in Section 1, HawkRAG's memory module is designed to achieve three key objectives: 1) length scalability, enabling efficient handling of long contexts; 2) retentiveness, ensuring the retention of crucial information from these contexts; and 3) instructiveness, providing useful clues that facilitate comprehensive retrieval. The first two objectives are met through specialized model designs, while the third is achieved via multi-stage, data-driven training.

**Memory Model Design:** The inference workflow in LLMs consists of two stages: (i) the prefill stage, where the input sequence is processed to generate key-value (KV) cache for each transformer layer; and (ii) the decoding stage, where the model sequentially generates tokens by utilizing and updating the KV cache.

In the prefill stage, let the input tensor $X \in \mathbb{R}^{n \times d} = \{x_1, \cdots, x_n\}$ consist of $n$ token embeddings, where $d$ is the model's hidden size. The input $X$ is processed by a transformer-based model $\Theta(\cdot)$, and the key-value cache $[\mathcal{K}, \mathcal{V}]$ are generated as follows:

$$\mathcal{K} = XW_{\mathcal{K}}, \quad \mathcal{V} = XW_{\mathcal{V}}, \tag{3}$$

where $W_{\mathcal{K}}$ and $W_{\mathcal{V}}$ are the weight matrices for the key and value projections, respectively. This attention mechanism is applied independently at each layer and for each attention head. For simplicity, we omit the layer and head indices in the equations.

In the decoding stage, let $\mathbf{t} \in \mathbb{R}^{t \times d}$ represent the new input tensor, where $t$ is the length of the newly input tokens. We compute the new key and value as:

$$\mathcal{K}_{\mathbf{t}} = \mathbf{t}W_{\mathcal{K}}, \quad \mathcal{V}_{\mathbf{t}} = \mathbf{t}W_{\mathcal{V}}. \tag{4}$$

The KV cache is then updated by concatenating the new key-value pairs with the previous ones:

$$\mathcal{K} \leftarrow \text{Concat}(\mathcal{K}, \mathcal{K}_{\mathbf{t}}), \quad \mathcal{V} \leftarrow \text{Concat}(\mathcal{V}, \mathcal{V}_{\mathbf{t}}). \tag{5}$$

Finally, the attention output is computed as:

$$Q_{\mathbf{t}} = \mathbf{t}W_Q, \quad A(Q, \mathcal{K}, \mathcal{V}) = \text{softmax}\left(\frac{Q_{\mathbf{t}}\mathcal{K}^T}{\sqrt{d}}\right)\mathcal{V}, \tag{6}$$

where $W_Q$ is the weight matrix for the query projection, and $A(\cdot)$ represents the attention function. For simplicity, we ignore other parts of the inference process.

*Light Global Memory.* The key-value cache computed during the prefill stage can be efficiently reused in the decoding stage. Thus, the key-value cache $[\mathcal{K}, \mathcal{V}]$ serves as the simplest form of global memory, denoted as $\theta_{\text{mem}} = [\mathcal{K}, \mathcal{V}]$. However, maintaining a full key-value cache for long contexts is computationally expensive and time-consuming. In this place, we first introduce a kind of baseline solution called *light global memory*, which directly takes

advantage of recent light long-context techniques, e.g., MInference [24] and SelfExtend [27]. Formally, they can be defined as $\theta_{\text{mem\_lite}} = v(\Theta(X \mid \theta))$, where $v(\cdot)$ represents the optimization techniques applied to the model.

While light global memory is easy to implement, empirical analysis in Section 3.4 demonstrates that it is inferior to the compact global memory introduced below. This is due to several factors: (1) it is constrained by the native context size of LLMs, limiting its adaptability to extremely long contexts; and (3) the use of sparse attention compromises semantic completeness. Besides, although light memory reduces parameters, it still consumes substantial GPU memory by maintaining the full length of the key-value cache

*Compact Global Memory.* We propose a flexible model architecture designed to facilitate efficient memory formation. The memory model progressively compresses the raw input tokens into a significantly smaller set of memory tokens in KV space, while preserving essential semantic information, resulting in compact global memory. Specifically, we introduce memory tokens $x^m$ to serve as the information carriers of global memory in LLMs. Suppose the LLM $\Theta(\cdot)$ has a working context window length of $l$. After each context window, we insert $k$ memory tokens, such that:

$$X = \{x_1, \cdots, x_l, x_1^m, \cdots, x_k^m, x_{l+1}, \cdots\}, \quad k \ll l. \tag{7}$$

For the memory tokens, we initialize a separate set of weight matrices, denoted as $W_{Q^m}$, $W_{\mathcal{K}^m}$, and $W_{\mathcal{V}^m}$, specifically for the purpose of memory formation. Let the memory tokens be denoted by $X^m$, and we compute the corresponding query, key, and value as follows:

$$Q^m = X^m W_{Q^m}, \quad \mathcal{K}^m = X^m W_{\mathcal{K}^m}, \quad \mathcal{V}^m = X^m W_{\mathcal{V}^m}, \tag{8}$$

$$A(Q, \mathcal{K}, \mathcal{V}) = \text{softmax}\left(\frac{[Q; Q^m]\tilde{\mathcal{K}}^T}{\sqrt{d}}\right)\tilde{\mathcal{V}}, \tag{9}$$

$$\tilde{\mathcal{K}} = [\mathcal{K}_{\text{cache}}^m; \mathcal{K}; \mathcal{K}^m], \quad \tilde{\mathcal{V}} = [\mathcal{V}_{\text{cache}}^m; \mathcal{V}; \mathcal{V}^m], \tag{10}$$

where $Q^m$, $\mathcal{K}^m$, and $\mathcal{V}^m$ are the query, key, and value for the memory tokens $X^m$. The terms $\mathcal{K}_{\text{cache}}^m$ and $\mathcal{V}_{\text{cache}}^m$ represent the KV cache for previously computed memory tokens.

In the prefill stage, after processing each context window, we generate new KV cache for the memory tokens, denoted as $[\mathcal{K}^m, \mathcal{V}^m]$. We update the previous memory token cache as follows:

$$\mathcal{K}_{\text{cache}}^m \leftarrow \text{Concat}(\mathcal{K}_{\text{cache}}^m, \mathcal{K}^m), \tag{11}$$

$$\mathcal{V}_{\text{cache}}^m \leftarrow \text{Concat}(\mathcal{V}_{\text{cache}}^m, \mathcal{V}^m). \tag{12}$$

Meanwhile, the KV cache $[\mathcal{K}, \mathcal{V}]$ for the regular tokens are discarded to reduce memory consumption. For compact global memory, we have $\theta_{\text{mem}} = [\mathcal{V}_{\text{cache}}^m, \mathcal{K}_{\text{cache}}^m]$. In our experiments, we typically select a compression ratio $\beta = l/k \in [4, 8, 16, 32, 64]$, resulting in an approximate $\beta\times$ reduction in GPU memory usage. Furthermore, since the number of memory tokens is much smaller than the number of raw tokens, LLMs can handle significantly longer contexts than their native context window would typically allow. For example, a 128K context LLM can process up to an 8M token context when a compression ratio of $\beta = 64$ is applied.

**Memory Model Training:** Since the memory model initializes a new set of parameters, we begin by training the memory model through pre-training. Following this, we perform supervised fine-tuning (SFT) using task-specific SFT data. Finally, we apply a small

set of SFT data labeled with preferences to perform preference alignment for the memory model.

*Pre-Training.* During the pre-training stage, the optimization goal is to enable the memory model to generate a global memory representation from raw input contexts. We only optimize the newly initialized weight matrices, $W_{Q^m}$, $W_{K^m}$, and $W_{V^m}$, while keeping the underlying LLM's parameters frozen. The model's objective is to predict the next token using the memory tokens and the current context. This can be expressed using a cross-entropy loss:

$$\mathcal{L}_{\text{pre}} = -\sum_{t=1}^{T} \log \mathcal{P}(x_t \mid x_{\text{cache}}^m, x_{1:t-1}), \tag{13}$$

where $x_{\text{cache}}^m$ represents the previously accumulated memory tokens, and $x$ represents the raw tokens. This loss encourages the model to maximize the probability of generating the correct next token based on the previous memory and the current raw context.

*Supervised Fine-Tuning.* In the SFT stage, the loss function is designed to help HawkRAG generate task-specific clues that can later guide the retrieval of relevant evidence. Here, the model is trained to minimize the difference between the generated output and the ground-truth outputs provided by the SFT dataset. The loss function is also a cross-entropy loss, but applied to task-specific data:

$$\mathcal{L}_{\text{SFT}} = -\sum_{t=1}^{T} \log \mathcal{P}(y_t \mid x_{\text{cache}}^m, q), \tag{14}$$

where $y$ represents the ground-truth task-specific output and $q$ is the query or task instruction. This loss ensures that HawkRAG learns to produce accurate clues based on the global memory. The SFT data is initially generated using strong LLMs and subsequently reviewed and refined by human annotators (see Appendix C for details). While the SFT data labels capture both LLM and human preferences regarding the answer clues, they do not directly reflect the quality of the final generated answers. To address this, we further optimize the memory module using a tailored optimization method which is introduced below.

*RLGF (Reinforcement Learning with Generation Feedback).* To further optimize the memory module for generating truly useful answer clues, the memory model is trained to align its outputs with preferred answer clues, selected based on their contributions to the overall end-to-end performance. The loss function is derived from a preference-based ranking loss, which encourages the model to prioritize outputs that lead to better evidence retrieval and final answer generation. This is defined as:

$$\mathcal{L}_{\text{RLGF}} = \sum (y^+, y^-) \max \left( 0, 1 - R(y^+) + R(y^-) \right), \tag{15}$$

where $R(y^+)$ and $R(y^-)$ represent the rewards assigned to the preferred and non-preferred outputs, respectively. This loss function drives the model to generate outputs that align more closely with the preferred answers, ensuring that the generated clues are both relevant and lead to improved evidence retrieval. As a result, the overall answer quality is enhanced. See Appendix C for details on the data construction for RLGF.

# 3 Experiment

In this section, we investigate the following research questions (RQ):

**RQ1**: *How does HawkRAG's performance compare to that of standard RAG systems, advanced RAG systems and long-context LLMs?*

**RQ2**: *Can HawkRAG effectively generalize beyond straightforward QA tasks to handle non-QA tasks and complex QA tasks involving long contexts and diverse domains?*

**RQ3**: *Are HawkRAG's model designs and optimization strategies well-justified and appropriately selected?*

**RQ4**: *How do HawkRAG's inference time efficiency and GPU memory usage compare to baseline methods?*

## 3.1 Dataset

To explore **RQ1** and **RQ2**, we evaluate HawkRAG and baselines using LongBench and InfiniteBench, two widely recognized benchmarks for long-context tasks [4, 55], which include the following tasks: (1) Single-Doc QA: NarrativeQA [29], Qasper [9], and MultiFieldQA [4]. (2) Multi-Doc QA: HotpotQA [53], 2WikiMQA [19], and MuSiQue [47]. (3) Non-QA tasks: GovReport [20], En.SUM [55] and MultiNews [14]. (4) Long-book QA: En.QA [55]. For summarization tasks, we use the task instruct as a fake query.

To further address **RQ2**, we evaluate HawkRAG across a broader range of real-world scenarios by introducing the UltraDomain benchmark, which consists of 20 datasets featuring long contexts and high-level queries across various specialized domains. Many of these tasks require a deep understanding of the entire context and the ability to synthesize multiple pieces of information to generate accurate answers. Additional details about UltraDomain can be found in Appendix D. More information on the training datasets and statistic information of all datasets can be found in Appendix C.

## 3.2 Baselines

We compare HawkRAG against three types of baselines: **(1) Using Full Context**: In this setting, we feed the full context into long LLMs, referred to as **Full**. For the main experiments, we utilize LLMs with a 128K context length, allowing to process all evaluation data samples without truncation. In addition to directly processing the full context, we explore two recent techniques that optimize context pre-filling for comparison: **MInference** [24], which applies strategic sparse attention to accelerate the pre-filling process, and **SelfExtend** [27], which constructs bi-level hierarchical attention to expand the original LLM's context length. **(2) Standard RAG with Alternative Retrieval Methods**: BGE-M3 [7]: A widely used retrieval model that has proven effective across many applications. **Stella-en-1.5B-v5**[12]: A state-of-the-art retrieval method that ranks in the top 3 on the MTEB leaderboard at the time of writing this paper. **Jina-emb-v3** [45]: A newly released frontier multilingual retrieval model, which claims to perform well in various scenarios, particularly in RAG tasks. **(3) Advanced RAG Methods**: **RQ-RAG** [6]: RQ-RAG prompts LLMs to refine the input query into several sub-queries that are more effective for retrieval by explicit rewriting, decomposition, and disambiguation. The supporting passages are retrieved using both the original and refined queries. **HyDE** [15]: Directly prompts LLMs to generate hypothetical documents based solely on the query, and then retrieves relevant passages using these documents. The final answer is generated based

Table 1: Main experiment results. Best results are in bold, second-best are underlined, and "†" indicates performance surpasses all baselines in a t-test at $p < 0.05$. Evaluation metrics for all datasets are in Appendix C.

| Dataset | nar | qas | mul | mus | 2wiki | hot | news | gov | en.sum | en.qa | fin | legal | misc | ave. |
|---|---|---|---|---|---|---|---|---|---|---|---|---|---|---|
| | | | | | **LongBench** | | | | **InfBench** | | | **UltraDomain** | | |
| Full | 21.4 | 39.4 | 51.5 | 28.2 | 38.1 | 48.1 | 24.9 | 32.6 | 13.0 | 15.2 | 47.8 | 46.5 | 48.7 | 35.0 |
| Mnference | 20.7 | 39.0 | 50.8 | 27.4 | 35.9 | 46.2 | 24.8 | 32.2 | 13.3 | 12.1 | 44.7 | 39.8 | 46.3 | 33.3 |
| SelfExtend | 19.6 | 37.8 | 47.4 | 22.7 | 37.2 | 42.0 | 21.4 | 29.1 | 11.1 | 9.3 | 41.2 | 37.9 | 34.1 | 30.1 |
| BGE-M3 | 20.3 | 33.0 | 44.3 | 21.1 | 35.4 | 42.1 | 17.7 | 19.8 | 9.6 | 16.3 | 41.7 | 41.2 | 43.7 | 29.7 |
| Stella-v5 | 13.7 | 32.4 | 43.5 | 21.0 | 35.6 | 40.6 | 20.3 | 18.2 | 10.0 | 19.5 | 42.8 | 35.1 | 43.9 | 29.0 |
| Jina-emb-v3 | 15.9 | 34.7 | 42.8 | 17.8 | 33.1 | 41.8 | 21.9 | 25.2 | 11.3 | 18.7 | 41.8 | 37.1 | 43.8 | 29.7 |
| GraphRAG | 16.2 | 36.3 | 45.4 | 19.3 | 37.5 | 38.0 | 18.4 | 25.6 | 10.8 | 13.5 | 39.9 | 39.6 | 41.7 | 29.4 |
| RQ-RAG | 19.6 | 34.1 | 46.5 | 21.9 | 36.1 | 41.7 | 20.1 | 18.6 | 10.4 | 16.1 | 41.8 | 40.9 | 43.2 | 30.1 |
| HyDE | 18.7 | 36.0 | 47.5 | 20.5 | 36.8 | 42.7 | - | - | - | 19.6 | 43.1 | 41.6 | 44.2 | - |
| HawkRAG | **27.5**† | **43.9**† | **52.2**† | **33.9**† | **54.1**† | **54.8**† | **26.3**† | **32.9**† | **15.7**† | **22.9**† | **51.5**† | **51.0**† | **55.6**† | **40.2** |

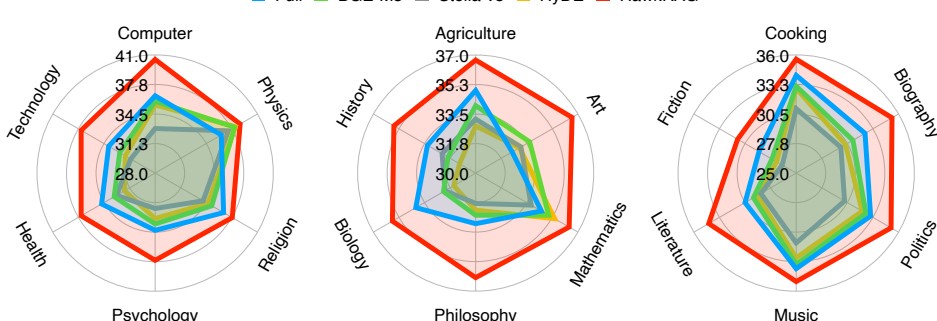

Figure 3: Experiment results on the **UltraDomain** benchmark. These datasets feature contexts of up to one million tokens, covering a wide range of subjects. See more details about the benchmark in Appendix D.

on the retrieved passages. **GraphRAG** [13]: A graph-based RAG framework that transforms unstructured data into graph structures, enabling the system to perform more complex question-answering tasks based on graph-based information retrieval.

In the main experiments, the memory model is trained on Mistral-7B-Instruct-v0.2-32K. By default, HawkRAG uses the underlying LLM of memory model as the generator. But Mistral's 32K context window is insufficient for most evaluation dataset contexts. To avoid context truncation, we use Phi-3-mini-128K-instruct [1] as the generator for HawkRAG and all baseline methods except for SelfExtend, which is specifically designed to enable LLMs to process contexts much longer than their native window. SelfExtend utilizes Phi-3-mini-4K-instruct as the generator and adjusts its effective context window according to the maximum context length required by different tasks. For GraphRAG, we utilize the OpenAI's GPT-4o API for all requests during both the indexing and searching processes. The results from GraphRAG's global search setting are extracted and used as the grounding evidence for answer generation[2]. See Appendix A for more implementation details.

### 3.3 Main Experiments

To address **RQ1** and **RQ2**, we compare HawkRAG against all baseline models across three benchmarks, as presented in Table 1. The

experimental results demonstrate that HawkRAG consistently outperforms all baselines across the evaluated datasets:

**First**, while RAG is a promising solution for long-context tasks, using long LLMs that handle the full context length often yields better performance (Full vs. other baselines). In contrast, HawkRAG significantly surpasses the performance of long LLMs, highlighting its superior ability to process long-context tasks. **Second**, for straightforward QA tasks from LongBench and InfiniteBench, HawkRAG outperforms all baselines, showing its effectiveness in standard RAG scenarios with explicit information needs. Its memory-generated clues allow for more accurate evidence retrieval from long contexts. In complex QA tasks (e.g., financial and legal), HawkRAG achieves notable improvements, demonstrating its capability to handle complex, long-context challenges. **Third**, while traditional RAG methods often struggle with non-QA tasks that lack explicit queries—such as summarization tasks (e.g., MultiNews, GovReport, and En.SUM)—HawkRAG excels. It efficiently extracts key points from the input context and retrieves additional details to generate comprehensive summaries.

To further address **RQ2**, we evaluate HawkRAG on the remaining 18 diverse datasets from UltraDomain, where most input contexts exceed the generator's context limit (e.g., 128K tokens). The results, presented in Figure 3, lead to the following conclusions: **First**, HawkRAG consistently outperforms all baselines across all datasets,

---

[2]https://microsoft.github.io/graphrag/posts/query/0-global_search/

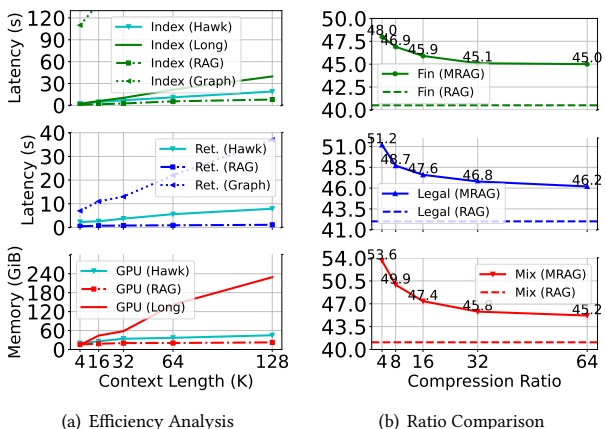

Figure 4: Ablation study. Figure (a) and (b) show the performance of different LLMs and optimization strategies. The *Pretrain*, *SFT*, and *RLGF* settings refer to the training stages. The *Light* setting uses the light memory model, introduced in Section 2.3. The *Zero* setting uses native LLMs without prior training. Figure (c) shows the outcomes of using different models as the generator.

Figure 5: Analysis on the model efficiency (left) and the impact of the choice of the compression ratio $\beta$ (right).

(a) Efficiency Analysis    (b) Ratio Comparison

demonstrating strong domain generalization capabilities. **Second**, directly inputting the full context into LLMs generally yields better performance compared to standard RAG methods, revealing that RAG systems struggle with high-level queries and locating relevant evidence. **Third**, HawkRAG surpasses the performance of directly using the full context, illustrating its ability to effectively process super-long contexts and address complex tasks.

**In summary**, HawkRAG consistently outperforms standard and advanced RAG systems, as well as long LLMs. It generalizes well beyond straightforward QA tasks, effectively handling non-QA tasks and complex QA tasks. Its advantages, driven by global memory-enhanced retrieval, are especially evident in scenarios where standard RAG systems face challenges.

## 3.4 Ablation Study

To address **RQ3**, we conduct comprehensive ablation studies:

**1) Model design and optimization strategy**: We first compare two memory model design options: *light memory* and *compact memory* (see Section 2.3). Additionally, we evaluate the performance of the HawkRAG pipeline using memory models at various stages of training. This includes a zero-shot evaluation, where the foundation model is directly applied to HawkRAG, as well as evaluations following pretraining, supervised fine-tuning (SFT), and reinforcement learning with generation feedback (RLGF). The results, shown in Figure 4 (a) and (b), indicate that each technical design contributes

uniquely to HawkRAG's overall effectiveness. Removing any of these designs results in performance degradation, validating the necessity and impact of HawkRAG's technical components.

**2) Foundation model choice**: To assess the impact of the foundation model, we replace the underlying LLM of HawkRAG's memory model with Qwen2-7B-instruct, which has a native context window of 128K tokens [52]. By comparing Figure 4 (a) and (b), we observe that utilizing either model as the foundation for HawkRAG's memory module results in consistent performance improvements. This demonstrates that HawkRAG's memory model design is robust and adaptable across a wide range of LLMs.

**3) Alternative generators**: We evaluate HawkRAG's effectiveness with three different generators: Llama3.1-8B-inst-128K, Mistral-7B-inst-v0.2-32K, and Phi-3-mini-128K. As shown in Figure 4 (c), HawkRAG consistently outperforms the direct use of long LLMs, with the performance gap widening as the task context exceeds the LLM's native context length. This indicates that HawkRAG can significantly enhance task performance when integrated with various LLMs as generators.

**4) Impact of compression rate**: As discussed in Section 2.3, the compression rate $\beta$ during compact memory formation affects both efficiency and effectiveness. A smaller $\beta$ retains richer semantics but requires more KV cache, while a larger $\beta$ improves efficiency but reduces semantic richness. We experimented with $\beta \in [4, 8, 16, 32, 64]$, and the results, shown in Figure 5 (b), indicate that as $\beta$ increases, performance declines but stabilizes at $\beta = 32$. Despite higher compression, HawkRAG consistently captures key information and outperforms the standard RAG pipeline across all values of $\beta$.

**In summary**, the ablation studies confirm the effectiveness of HawkRAG's technical designs and model choices, demonstrating that its architecture is well-motivated and robustly designed.

## 3.5 Efficiency Analysis

To address **RQ4**, Figure 5(a) compares model efficiency[3]. Key observations include: (1) **Indexing latency analysis** (top): Standard RAG quickly indexes long inputs due to its simpler process, while HawkRAG is slower due to the global memory formation. However, it remains more efficient than long LLMs' pre-filling, thanks to its optimized memory model. GraphRAG is the slowest, heavily reliant on GPT-4 APIs. (2) **Retrieval latency analysis** (middle): Standard RAG retrieves efficiently using vector databases (e.g., FAISS [28]),

---

[3]We randomly selected 5 samples with 128K context lengths from the UltraDomain benchmark, truncating the context into shorter segments to test various methods under the same configuration.

**Table 2: Case study on the Legal dataset. Predicted answers that overlap with the ground-truth answers are marked in teal.**

**Query**: What is the significance of the Outside Date mentioned in the agreement?    **Context**: A Legal Contract (56.4K tokens)
**Ground-truth target**: The Outside Date is the deadline by which the Plan must become effective, or else the Agreement will terminate automatically. It is set as October 5, 2020, at 11:59 p.m. Eastern Time.

**Standard RAG**: The Outside Date is significant as it is a date where both parties have agreed in advance that if the merger or acquisition has not yet completed either side. It is set as October 5, 2020. (F1-Score: 0.36)

**Clues #1**: Definition of the "Outside Date" in the agreement    **Clues #2**: "Outside Date" means October 5, 2020 at 11:59 p.m. Eastern Time.
**HawkRAG**: The Outside Date mentioned in the agreement is October 5, 2020, at 11:59 p.m. Eastern Time. It is a significant date in the context of the agreement because it is the deadline for the Plan to become effective. If the Plan has not become effective by this date, certain parties may have the right to terminate the agreement. (F1-Score: 0.83)

while HawkRAG is slower as it generates retrieval clues but still outperforms GraphRAG. (3) **GPU memory consumption analysis** (bottom): Both HawkRAG and standard RAG process 128K contexts with under 60 GiB of GPU memory, whereas long LLMs require substantially more due to the large key-value cache. **In summary**, HawkRAG maintains a balanced time and memory efficiency. While it is slower than standard RAG, it outperforms advanced RAG methods and long LLMs in both time and memory efficiency.

### 3.6 Case Study

In Table 2, we present an example processed by HawkRAG. The input query pertains to the high-level understanding of the term "Outside Date" within the input context, a legal contract consisting of 56.6K tokens. The standard RAG system searches for evidence solely based on the input query, in which the semantics of "significance of the Outside Date" is not explicitly present. Therefore, direct semantic connections with the expected supporting evidence are difficult to establish. As a result, the standard RAG system generates answers that provide a general definition of the term "Outside Date" rather than its "significance" regarding this legal contract. Our HawkRAG, on the other hand, benefits from the global perception of the entire input context. It can evoke several clues that bridge the semantic gap between the expected supporting evidence and the input query. By leveraging these clue texts, we can more accurately locate the relevant evidence passages, leading to a more comprehensive and precise response.

## 4 Related Work

**Long Context:** Handling long contexts is a fundamental issue for LLMs. The most straightforward approach is to train LLMs on long text sequences, giving them a native ability to handle extended contexts [1, 5, 10, 38]. However, this is very expensive, as computational costs increase exponentially with longer contexts. As a result, researchers focus on improving attention efficiency [3, 8, 10, 23]. Additionally, Liu et al. [33] highlight that LLM performance may degrade when the target answer is located in the middle of the context. To address this, various works explore data augmentation, attention reweighting, and data re-organization [17, 32, 33, 50].

Another approach involves compressing the input through strategies like sliding windows, context compression, and summarization [25, 30, 42, 51, 54]. With the rapid development of long-context processing, context windows for LLMs have expanded significantly, from 4K tokens (e.g., Llama-2[46]) to 128K tokens (e.g., Phi-3, GPT-4[1, 38]). Recent advancements even allow LLMs to extend their context window to 1 million tokens [17]. Additionally, RAG has become a common solution for long-context challenges, using retrieval to find precise evidence within large inputs [51].

**RAG:** Retrieval-augmented generation (RAG) was initially introduced by Lewis et al. [31], defining a retrieval process that assists language models in handling knowledge-intensive tasks. Subsequent RAG research has focused on two areas: improving retrieval quality, which sets the upper bound for final generation quality [16, 37, 48, 49], and enhancing the use of retrieved passages for increased relevance and flexible access [21, 26, 39].

With recent advancements in LLMs, incorporating RAG into LLM-based systems has become popular, inspiring numerous applications [43]. As a result, there has been a growing call for more general-purpose RAG systems [56]. However, the standard RAG pipeline faces inherent limitations and struggles to generalize effectively in complex tasks involving implicit information needs [16].

To expand RAG's applicability, recent works have proposed modifying the RAG pipeline with tailored approaches. For instance, HyDE generates a hypothetical document from the query, which is used to retrieve relevant evidence [15], while RQ-RAG rewrites the query into simpler forms to improve retrieval [6]. However, both rely solely on the model's internal knowledge, limiting their effectiveness for domain-specific tasks. GraphRAG [13] constructs a knowledge graph to assist retrieval, but its static graph construction is difficult to optimize. Other methods [6, 18, 40] also fail to achieve a comprehensive understanding of the input context, leading to incomplete semantic comprehension.

## 5 Conclusion

In this paper, we tackle long-context processing using global memory-enhanced retrieval by introducing HawkRAG, a framework that builds a global memory from the entire context. When presented with a task, HawkRAG generates draft answers that, although lacking in detail, effectively guide the retrieval of relevant evidence for more accurate final response generation. By leveraging these clues, HawkRAG identifies precise information within the long context, improving overall answer quality. Extensive experiments on two long-context benchmarks and various real-world applications demonstrate that HawkRAG significantly outperforms standard RAG systems, advanced RAG systems and long LLMs. HawkRAG excels in tasks requiring high-level information aggregation, while also offering notable advantages in traditional tasks commonly handled by previous RAG systems, expanding the potential and applicability of RAG to a broader range of scenarios.

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

## A Implementation Details

For pre-training the memory model, we sample text spans from the RedPajama [44] dataset to create a training set of 2 billion tokens. The memory context window size is set to 2048, and during training, we randomly select a compression ratio $\beta \in [4, 8, 16, 32, 64]$ for each context window. The model is trained for 1 epoch with a batch size of 8 and a learning rate of 5e-5.

For supervised fine-tuning (SFT), we build an SFT dataset consisting of 17,116 samples. In this stage, the model is trained for 2 epochs with a batch size of 8 and a learning rate of 1e-5. The lengths of the SFT samples range from 4K to 64K tokens.

During RLGF optimization, we sample 2,000 instances from the SFT training dataset and rank the generated clue answers, categorizing them into preferred and rejected based on their contributions to the overall end-to-end performance. The data construction process can refer to Appendix C.

During the memory module training, we keep the underlying model's parameters frozen and train only the newly initialized parameters of the memory model, avoiding the resource-intensive

process of full parameter fine-tuning. The size of the newly initialized parameters varies depending on the underlying LLM. For instance, with Qwen2-7B-instruct, the newly initialized parameters are approximately 1.1 billion.

For the light global memory setting, we utilize SelfExtend [27] to extend the LLMs' context window to the maximum length required for each specific task. Additionally, we apply MInference [24] to accelerate the prefill process.

For the main experiments, we set the compression ratio to $\beta = 4$. For HawkRAG, RQ-RAG and HyDE, we use BGE-M3 [7] as the retriever and set the hit number to 3. We use the semantic-text-splitter tool to chunk the long context with a maximum length of 512. For HawkRAG and all baselines, we use the same task prompts provided by the official repositories of the corresponding benchmarks[4]. We also use the same generation hyper-parameters (varying by task) for HawkRAG and all baseline models.

All training and evaluation were conducted using 8 NVIDIA A800-80G GPUs.

## B  Prompts

For memory formation, we use the prompt in Table 3. For the memory clue generation, we use the prompt in Table 4 for QA tasks and we use the prompt in Table 5 for summary tasks. For the evaluation tasks, we use the provided task prompts in the corresponding GitHub repositories.

## C  More details of Dataset Construction

To construct the SFT training set, we first collect long contexts from novels, academic papers, news, financial reports, and legal contracts. The collection of novels, academic papers, and news comes from the training datasets of NarrativeQA, Qasper, and HotpotQA. The legal contracts are sourced from this repository, and the financial reports are from this repository. We then sample long contexts of up to 80K tokens and use strong LLMs (e.g., GPT-4 128K) to generate high-level, insightful question-answer pairs. After quality review, we selected 20,000 samples and prompted the same LLMs to generate answer clues that bridge the gap between the query and the long context. During this process, the LLMs were provided with the query, the long context, and the answer, enabling them to utilize both priori and posteriori knowledge to generate the answer clues more effectively. These clues were then inspected for quality through human review, resulting in 17,116 SFT training samples. Six graduate students participated in the inspection, with each sample reviewed by at least three students. Samples tagged as *discard* more than twice were excluded from the final dataset.

For the RLGF training set, we selected 2,000 samples from the SFT dataset, filtering for those with more than five answer clues. For each clue, we retrieved the top-3 evidence. We then greedily evaluated the performance of all combinations of three or more clues and identified the best-performing combination as the preferred answer and the worst-performing combination as the rejected answer.

## D  More details of UltraDomain

We begin constructing the UltraDomain benchmark by leveraging contexts from datasets representing specific areas of knowledge, focusing on two specialized datasets. The first is the *Fin* dataset, derived from financial reports, which tests HawkRAG's ability to process and interpret complex financial data, ensuring it can manage the intricacies of financial language and reporting. The second is the *Leg* dataset, composed of legal contracts, which challenges HawkRAG to comprehend and navigate the precise, nuanced language of legal documents.

In addition to these specialized datasets, we collected a diverse set of 428 college textbooks covering 18 distinct domains, including natural sciences, humanities, and social sciences[5]. These textbooks are used to evaluate HawkRAG's versatility and adaptability across a broad range of topics, including those unrelated to finance and law. By assessing HawkRAG on these varied contexts, we gain insights into its potential for broader applications beyond specific domains. We also created a *Misc* dataset, comprising mixed contexts from the specialized datasets. This dataset is designed to assess HawkRAG's ability to generalize across different types of contexts.

Specifically, we sampled text spans up to 128K tokens in length and fed them into GPT-4, prompting it to generate high-level question-answer pairs that require a comprehensive understanding of the full context. Six graduate students manually reviewed the generated QA pairs by: (1) selecting questions that are not directly searchable, and (2) evaluating the quality of the generated answers. This process yielded a total of 3,240 evaluation samples.

Statistical details of the UltraDomain benchmark are provided in Table 7 and Table 8. Together, these datasets form a rigorous benchmark for evaluating HawkRAG's effectiveness in both domain-specific tasks and broader, cross-disciplinary applications. Example cases from UltraDomain are shown in Table 9. A guidebook for constructing UltraDomain will be released upon publication.

---

[4]LongBench: https://github.com/THUDM/LongBench, InfiniteBench: https://github.com/OpenBMB/InfiniteBench

[5]https://huggingface.co/datasets/P1ayer-1/books-3-textbooks

**Table 3: Prompt for Global Memory Formation.**

You are provided with a long article. Read the article carefully. After reading, you will be asked
to perform specific tasks based on the content of the article.
### Article Content:
- {context}
### Instructions:
- The article ends here.
- Follow the instructions provided to complete the tasks.

**Table 4: Prompt for Generating Answer Clues for QA Tasks.**

You are given a question related to the article. To answer it effectively, you need to recall specific details from the article.
Your task is to extract specific clue texts from the article, or generate clues questions that are relevant to the question.
### Question: {question}
### Instructions:
1. You have a general understanding of the article. Your task is to generate one or more specific clue text spans or clue
questions that will help in searching for supporting evidence within the article.
2. The clue text are in the form of text spans that will assist in answering the question.
3. The clues questions are in the form of precise surrogate questions that clarify the original question.
4. Only output the clues. If there are multiple clues, separate them with a newline.

**Table 5: Prompt for Summarization Task.**

Your task is to create a concise summary of the long article by listing its key points. Each key
point should be listed on a new line and numbered sequentially.
### Requirements:
- The key points should be brief and focus on the main ideas or events.
- Ensure that each key point captures the most critical and relevant information from the article.
- Maintain clarity and coherence, making sure the summary effectively conveys the essence of the article.

**Table 6: Experiment results on UltraDomain. The evaluation metric is the F1-score, with the best results highlighted in bold and the second-best results underlined. The upward arrow ↑ indicates the improvement over the second-best results. ave($|C|$) refers to the average context length, counted in thousands of tokens (K).**

| UltraDomain | Full | BGE-M3 | Stella-v5 | HyDE | HawkRAG | ave($|C|$) (K) |
|---|---|---|---|---|---|---|
| Biology | 34.1 | 32.2 | 32.1 | 31.5 | **35.7** ↑1.6 | 125.2 |
| Religion | 36.7 | 35.2 | 34.1 | 34.7 | **37.8** ↑1.1 | 131.4 |
| Computer | 36.5 | 35.9 | 32.9 | 35.5 | **40.5** ↑4.0 | 215.9 |
| Fiction | 29.0 | 27.6 | 26.5 | 27.1 | **31.3** ↑2.3 | 137.7 |
| Literature | 30.5 | 29.6 | 28.8 | 29.2 | **34.4** ↑3.9 | 129.4 |
| History | 33.3 | 31.9 | 32.3 | 31.1 | **35.6** ↑2.3 | 195.2 |
| Biography | 32.4 | 31.1 | 29.8 | 30.3 | **35.3** ↑2.9 | 163.5 |
| Physics | 36.4 | 38.1 | 37.3 | 38.2 | **38.8** ↑0.6 | 105.8 |
| Music | 33.9 | 33.5 | 31.5 | 32.9 | **35.1** ↑1.2 | 168.7 |
| Art | 32.5 | 33.7 | 33.1 | 33.0 | **36.6** ↑2.9 | 129.0 |
| Mathematics | 34.5 | 35.0 | 33.8 | 35.4 | **36.4** ↑1.0 | 198.0 |
| Health | 34.8 | 33.2 | 32.9 | 31.9 | **37.4** ↑2.6 | 134.9 |
| Psychology | 34.3 | 33.6 | 31.9 | 33.0 | **37.6** ↑3.3 | 150.1 |
| Technology | 33.9 | 32.5 | 31.1 | 31.8 | **37.4** ↑3.5 | 144.0 |
| Politics | 33.0 | 32.5 | 30.2 | 32.1 | **35.2** ↑2.2 | 139.6 |
| Cooking | 34.1 | 33.1 | 31.0 | 32.9 | **35.6** ↑1.5 | 156.1 |
| Agriculture | 34.9 | 34.0 | 33.2 | 32.8 | **36.7** ↑1.8 | 151.0 |
| Philosophy | 33.0 | 32.5 | 31.8 | 32.2 | **36.2** ↑3.2 | 135.7 |
| Average | 33.8 | 33.0 | 31.9 | 32.5 | **36.2** ↑2.4 | 150.6 |

**Table 7: Statistical information of the datasets utilized in this paper.**

| Dataset | Narrative | Qasper | MultiField | Hotpot | MuSiQue | 2Wiki |
|---|---|---|---|---|---|---|
| Num of Samples | 200 | 200 | 150 | 200 | 200 | 200 |
| Ave. Length | 18,409 | 3,619 | 4,559 | 9,151 | 11,214 | 4,887 |
| Metric | F1 | F1 | F1 | F1 | F1 | F1 |
| Dataset | GovReport | MultiNews | En.Sum | En.QA | Fin | Legal |
| Num of Samples | 200 | 200 | 103 | 351 | 345 | 438 |
| Ave. Length | 8,734 | 2,113 | 171,500 | 192,600 | 40,625 | 51,413 |
| Metric | Rouge-L | Rouge-L | F1 | Rouge-L | F1 | F1 |

**Table 8: Statistical information of the out-of-domain evaluation datasets utilized in this paper.**

| Dataset | Num | $\max(|C|)$ | $\min(|C|)$ | $\text{ave}(|C|)$ | $\text{ave}(|Q|)$ | $\text{ave}(|\mathcal{A}|)$ |
|---|---|---|---|---|---|---|
| Technology | 240 | 306,073 | 44,549 | 144029.7 | 14.4 | 40.2 |
| Biology | 220 | 257,644 | 39,218 | 125284.9 | 16.8 | 49.1 |
| Religion | 220 | 1,071,342 | 34,257 | 131424.8 | 17.4 | 54.2 |
| Fiction | 220 | 564,980 | 44,057 | 137689.7 | 16.2 | 43.6 |
| Psychology | 200 | 571,725 | 37,988 | 150119.5 | 16.7 | 46.5 |
| Music | 200 | 381,043 | 51,517 | 168672.9 | 17.5 | 49.7 |
| Art | 200 | 305,001 | 32,793 | 128961.2 | 17.8 | 52.2 |
| Philosophy | 200 | 678,553 | 38,729 | 135682.7 | 17.2 | 51.0 |
| Health | 180 | 289,258 | 50,600 | 135902.0 | 16.2 | 48.2 |
| History | 180 | 688,074 | 53,277 | 195265.0 | 17.9 | 51.0 |
| Literature | 180 | 534,836 | 33,043 | 129363.7 | 16.9 | 47.0 |
| Biography | 180 | 408,969 | 45,052 | 163522.3 | 18.0 | 52.0 |
| Politics | 180 | 387,157 | 49,853 | 139624.3 | 17.9 | 54.9 |
| Mathematics | 160 | 726,144 | 60,936 | 197924.6 | 16.7 | 47.6 |
| Physics | 160 | 226,811 | 36,717 | 105805.6 | 14.8 | 54.2 |
| Cooking | 120 | 466,885 | 58,360 | 156139.2 | 16.5 | 46.6 |
| Agriculture | 100 | 385,915 | 76,581 | 150969.6 | 15.6 | 45.9 |
| Computer | 100 | 437,070 | 51,704 | 215929.5 | 14.3 | 39.8 |
| Total | 3,240 | 1,071,342 | 32,793 | 150684.0 | 16.6 | 48.5 |

**Table 9: Data cases of the domain data in UltraDomain**

| Domain | Book | Length | Query | Answer |
|---|---|---|---|---|
| mathematics | Lie Groups | 726K | What is Schur Orthogonality and why is it important in the representation theory of compact groups? | Schur Orthogonality states that if $(\pi_1, V_1)$ and $(\pi_2, V_2)$ are irreducible representations of a compact group G, then every matrix coefficient of $\pi_1$ is orthogonal in $L_2(G)$ to every matrix coefficient of $\pi_2$, unless the representations are isomorphic. This is crucial as it provides an orthonormal basis for L2(G) in terms of the matrix coefficients of ireducible representations. |
| biology | Butterflies | 189K | How does the book "Butterflies" utilize color photography and reproduction techniques? | The book "Butterflies" utilizes the latest methods of color photography and reproduction to portray the plants and animals in the full beauty of their natural colors, enhancing the visual appeal and educational value of the content. |
| history | Exemplary Women of Early China | 251K | How does the _Lienü zhuan_ reflect the historical context of the Former Han dynasty? | The _Lienü zhuan_ reflects the historical context of the Former Han dynasty by addressing the resurgence of consort power at court, which provided an incentive for Confucian thinkers to focus on shaping women's morals and their impact on dynastic health through didactic materials and moral education. |
| fiction | Hangsaman | 103K | What is the central theme of "Hangsaman"? | The central theme of "Hangsaman" is the exploration of consciousness and the development of an adult identity, particularly through the experiences of the protagonist, Natalie Waite, as she navigates the complexities of family dynamics, college life, and her own psychological struggles. |
| physics | Gravity | 37K | How does the Unified Field Theory attempt to reconcile gravity with other fundamental forces? | The Unified Field Theory, pursued by Einstein, aims to find a single, comprehensive framework that describes both gravity and electromagnetic forces (and potentially other fundamental forces) in a unified manner, suggesting that these forces may have a common underlying basis or origin. |

