# OpenReview forum: "Memory Never Fades: Boosting Long Context Processing with Global Memory-Enhanced Retrieval Augmentation"
_ACM.org/TheWebConf/2025/Conference — WWW 2025 Poster_

### Official Review · Reviewer_USmj · 2024-11-07

**Novelty:** 5
**Technical Quality:** 4

**Review:**

Pros:
This work focuses on the problem of long text in large language models. If we directly use the method of increasing the input window, the cost is relatively high and it will affect the performance. Therefore, this paper adopts the method of retrieval enhancement to solve the problem of long text input. In addition, we use a large model to get a draft answer based on the global information first, and then help the searcher retrieve useful results based on the draft answer. Moreover,  I think the design of Memory Model Design is more interesting.
Cons:
1. In the field of long context, some people have used methods similar to obtaining an initial answer or summarizing based on global information, such as [1]. In the field of non-long context, some people have used the point of obtaining a draft answer first and then retrieving useful information in RAG, such as [2, 3]. I don’t think this work is different from the listed works in terms of innovation.

[1] Zhao, Q., Wang, R., Cen, Y., Zha, D., Tan, S., Dong, Y., & Tang, J. (2024). LongRAG: A Dual-Perspective Retrieval-Augmented Generation Paradigm for Long-Context Question Answering. arXiv preprint arXiv:2410.18050.

[2] Zhang, H., Zhang, R., Guo, J., de Rijke, M., Fan, Y., & Cheng, X. (2024, July). Are Large Language Models Good at Utility Judgments?. In Proceedings of the 47th International ACM SIGIR Conference on Research and Development in Information Retrieval (pp. 1941-1951).

[3] Wang, D., Huang, Q., Jackson, M., & Gao, J. (2024). Retrieve What You Need: A Mutual Learning Framework for Open-domain Question Answering. Transactions of the Association for Computational Linguistics, 12, 247-263.

2. LongRAG, which is close to this work, is not used as a baseline. I think the experimental analysis of this work may be incomplete.

3. The analytical experiments are incomplete. The paper only conducts analytical experiments on the types of memory models and the length of  long context but does not test each module of HawkRAG, such as the impact of draft answers and no draft answers on retrieval results.

**Questions:**

1. LongRAG, which is close to this work, is not used as a baseline. I think the experimental analysis of this work may be incomplete.
2. The analytical experiments are incomplete. The paper only conducts analytical experiments on the types of memory models and the length of  long context but does not test each module of HawkRAG, such as the impact of draft answers and no draft answers on retrieval results.

**Reviewer Confidence:**

3: The reviewer is confident but not certain that the evaluation is correct

**Scope:**

4: The work is relevant to the Web and to the track, and is of broad interest to the community

---

### Official Review · Reviewer_GMKc · 2024-11-17

**Novelty:** 4
**Technical Quality:** 4

**Review:**

This paper introduces HawkRAG, a novel framework for long-context processing that leverages a global memory-enhanced retrieval augmentation system. It features a dual-system architecture to efficiently handle long contexts, retain crucial information, and generate useful clues for presented tasks. The paper demonstrates HawkRAG's superiority in processing long-context tasks compared to baseline methods.

**Questions:**

1.The abstract should be concise yet comprehensive, clearly stating the problem, methodology, and key findings. Currently, it lacks clarity in explaining the novelty of HawkRAG compared to traditional RAG methods.
2. Figure 2 should be further modified to facilitate the reader’s understanding. Specifically, the explanation of special symbols should be supplemented.
3. The authors should carefully polish the manuscript and correct the typos (e.g., Eq.(2)).
3. I suggest that adding a section discussing the strengths and weaknesses of existing RAG methods as well as how HawkRAG builds upon or differs from these approaches.
4. The manuscript should be edited for grammar, punctuation, and style to ensure clarity and readability.
5. Please review and update the references to ensure the consistency of citation formats in the use of full names or abbreviations for conference or journal names.

**Reviewer Confidence:**

2: The reviewer is willing to defend the evaluation, but it is likely that the reviewer did not understand parts of the paper

**Scope:**

4: The work is relevant to the Web and to the track, and is of broad interest to the community

---

### Official Review · Reviewer_2GH3 · 2024-11-19

**Novelty:** 3
**Technical Quality:** 3

**Review:**

This paper proposes a novel RAG framework called HawkRAG that aims to handle long-context processing through a dual-system architecture: a light but long-range system for creating global memory, and an expensive but expressive system for final answer generation. The framework is evaluated across various benchmarks including LongBench, InfiniteBench, and a new UltraDomain benchmark.

From a technical quality perspective, the paper demonstrates strong experimental evaluation and ablation studies but lacks rigorous theoretical analysis and statistical validation in key components. The theoretical foundation lacks rigorous analysis of both the memory module and RLGF optimization. Regarding clarity, while the dual-system architecture is presented coherently, several critical aspects remain unclear. The memory formation process and clue generation mechanisms need better explanation, and the RLGF training process is not well-articulated. In terms of originality, the paper shows moderate innovation. The concept of global memory-enhanced retrieval presents an interesting approach to long-context processing. However, many components build directly on existing techniques, and the RAG system modifications appear incremental rather than revolutionary. Regarding significance, while the problem being addressed is undoubtedly important for the field, the paper's impact is limited by several factors. The implementation complexity and computational requirements may hinder practical adoption, and the advantages over simpler alternatives are not convincingly demonstrated.

Strengths:

1. Addresses an important challenge in LLM development (long-context processing);
2. Comprehensive experimental evaluation across multiple benchmarks and thorough comparison against various baselines;
3. The paper demonstrates performance improvements over both standard RAG methods and long LLMs across multiple tasks, especially in non-QA tasks where traditional RAG methods struggle;

Weaknesses:

1. The analogy to human cognition in Section 1 feels forced and doesn't meaningfully connect to the technical implementation, particularly in explaining how the memory module actually mimics human memory processes.
2. The mechanism - using KV cache as global memory (Section 2.3) - appears to be an incremental extension of existing work on efficient attention mechanisms, with unclear differentiation from recent advances in memory-based transformers;
3. In the experimental comparisons (Section 3.2), the baseline configurations appear suboptimal;
4.The ablation study shows significant performance degradation with higher compression ratios (β>32), but this limitation is not adequately addressed in the discussion of scalability claims;
5. The fundamental mechanism of improvement appears to be quite straightforward: the system essentially uses an LLM to auto-generate multiple search cues for traditional RAG retrieval. While this does improve retrieval accuracy (as shown in Table 1 and Table 2), it's more of an engineering optimization than a fundamental advancement in RAG methodology. This is particularly evident in Section 2.2 where the paper describes how HawkRAG generates "draft answers" that serve as retrieval clues - this is essentially just using an LLM to create multiple reformulated queries, similar to but less transparent than existing query expansion techniques.

**Questions:**

1. How does the memory formation process fundamentally differ from existing work on efficient attention mechanisms and memory-based transformers?
2. How were the baseline RAG methods configured? Please provide detailed hyperparameter settings and justification for these choices.
3. Can you clarify the computational requirements and training costs compared to standard RAG approaches?
4. The paper mentions "human review" of training data - can you provide more details about this process and how potential biases were controlled?
5. Can you clarify the process of generating clues from the memory module? How do you ensure the generated clues are actually relevant to the task?

**Reviewer Confidence:**

3: The reviewer is confident but not certain that the evaluation is correct

**Scope:**

2: The connection to the Web is incidental, e.g., use of Web data or API

---

### Official Review · Reviewer_2yvR · 2024-11-26

**Novelty:** 4
**Technical Quality:** 5

**Review:**

## Pros

1. The **HawkRAG** framework effectively addresses the limitations of traditional RAG methods in handling long contexts through an innovative dual-system architecture and globally memory-augmented retrieval mechanism. By combining draft generation with complex generation, HawkRAG not only handles complex long-context tasks but also improves the quality and precision of the generated outputs.
2. The experiments are very solid, and the results outperform the baselines.
3. The readability of the paper is excellent.
4. The authors provide the code.

## Cons

1. The approach in this work is quite similar to the following two papers, and I would appreciate it if the authors could explain in detail the differences between their work and these two:
   - "HippoRAG: Neurobiologically Inspired Long-Term Memory for Large Language Models"
   - "MemoRAG: Moving towards Next-Gen RAG Via Memory-Inspired Knowledge Discovery"
2. I am curious why some of the works mentioned in the related work section were not included as part of the baselines.

**Questions:**

The cons mentioned above are my questions.

If the authors could address these two points, I would be happy to consider increasing the score ASAP.

**Ethics Review Flag:**

Yes

**Reviewer Confidence:**

2: The reviewer is willing to defend the evaluation, but it is likely that the reviewer did not understand parts of the paper

**Scope:**

4: The work is relevant to the Web and to the track, and is of broad interest to the community

---

### Official Review · Reviewer_k1fD · 2024-12-02

**Novelty:** 5
**Technical Quality:** 7

**Review:**

This paper introduces HawkRAG, which is empower by global memory-augmented retrieval. It employs a light but long-range system to create a global memory of the long context. Then, it leverages an expensive but expressive system to generates the final answer based on the retrieved information. The idea to build a global memory for question clarification and answer generation is interesting. However, the baselines did not include the latest strong models, which makes the conclusion not convincing enough.
Strengths:
1. The idea to build a global memory for long context is interesting.
2. The testbed used to evaluate the model is comprehensive, which includes long-bench, InfBench, and UltraDomain. The results also show the improvements overall different baselines.
3. The paper is clearly presented and easy to understand.

Weaknesses:
1. The experimental results on Table 1 show that the Full model outperforms all baselines on almost all datasets, which demonstrate that all these datasets are well addressed with current long-context LLMs. In fact, the rag baselines, i.e. GraphRAG, RQ-RAG, and HyDE, are all target on the retrieval step, which aims to obtain context from a large corpus. However, the hawkRAG targets on evidence extraction from the retrieved context. Thus, it would be better to compare with context-compression based rag methods as follows:
[1] LLMLingua-2: Data Distillation for Efficient and Faithful Task-Agnostic Prompt Compression, https://arxiv.org/abs/2403.12968
[2] Learning to Filter Context for Retrieval-Augmented Generation, https://arxiv.org/pdf/2311.08377
[3] RECOMP: IMPROVING RETRIEVAL-AUGMENTED LMS WITH COMPRESSION AND SELECTIVE AUGMENTATION, https://arxiv.org/pdf/2310.04408
2. The baselines are not strong enough to convincing the effectiveness of the proposed method. Overall, the HawkRAG firstly generate answer clue (also named pseudo answer or query expansions in other papers), then retrieve context for answer generating. In this process, the memory model acts as the in-memory query expansion model and the in-memory re-rank model, and it only takes the long context C as input rather than the whole corpus. In this way, the memory module is very similar to the Self-Inquiry and knowledge assimilation in ActiveRAG[1]. However, the paper did not cite it, let alone compare with it.
[1] ActiveRAG: Autonomously Knowledge Assimilation and Accommodation through Retrieval-Augmented Agents, https://arxiv.org/abs/2402.13547
3. The results on Figure 3 are somewhat weird that the 18 datasets have longer contexts exceed the generator’s context limit (e.g., 128K tokens). However, the results show that directly inputting the full context into LLMs generally yields better performance compared to standard RAG methods.

**Questions:**

1. Does the memory module process the whole corpus, e.g., the wikipedia dumps, or only the retrieved long documents?
2. What is main difference between the HawkRAG and ActiveRAG?

**Reviewer Confidence:**

4: The reviewer is certain that the evaluation is correct and very familiar with the relevant literature

**Scope:**

4: The work is relevant to the Web and to the track, and is of broad interest to the community